# Strategic classification made practical: reproduction

## Reproducibility Summary

**Scope of Reproducibility**

In this work, the paper Strategic Classification Made Practical[3] is evaluated through a reproduction study. The results from the reproduction examines if the claims made in the paper are valid. We could find two main claims that were made by the authors that we will attempt to reproduce. Those are as follows:

1. "We propose a novel learning framework for strategic classification that is practical, effective, and flexible. This allows for differentiation through strategic user responses, which supports end-to-end training."

2. "We propose several forms of regularization that encourage learned models to promote favorable social outcomes."

We interpret *practical, effective and flexible* as such that the model should work better on a variety of real life problems than their non-strategic counterpart.

**Methodology**

In this paper, the same code, datasets and hyperparameters were used as the original paper to reproduce the results. To further validate the claims from the original paper, we extended the original implementation to include an experiment that tests performance on a dataset containing both strategic (also referred to as gaming) and non-strategic users.

**Results**

The reproduction of the original paper as well as the extended implementation were successful. We were able to reproduce the original results and examine the performance of the proposed model in an environment where strategic and non-strategic users both present. Linear models seem to struggle with different proportions of strategic users, while the non-linear model (RNN) achieves good performance regardless of the proportion of strategic users.

**What was easy**

The codebase for the paper was available on GitHub which meant that we didn't have to start from scratch. They also provided us with the original data. The codebase also came with the original results from the authors which meant that comparing the results was easy.

**What was difficult**

Although the code was available, documentation of the code was quite sparse. Therefore, it was hard to figure out what each part of the code did and made it difficult to interpret what the results actually meant at certain stages.

**Communication with original authors**

The University of Amsterdam communicated before the course with the authors about the datasets. While working on the reproduction we sent one email about clarification of their method and to request a missing dataset.

Submitted to ML Reproducibility Challenge 2021 Fall Edition. Do not distribute.

# 1 Introduction

As consequential decisions such as loan approval and fraud detection are increasingly made by predictive machine learning systems, it is important to consider the weaknesses and vulnerabilities of these systems. Users may gain knowledge of the model and use this knowledge to modify their features to improve their outcomes. Therefore, a model must be resilient against strategic modification of features to classify these users properly.

This problem of classification while users strategically modify their features is referred to as strategic classification, and it is the main subject of the paper Strategic Classification Made Practical[3]. In this paper, a novel framework is proposed that claims to be more practical and flexible than previous methods, along with novel methods to improve social outcomes in automated decision-making. This work will examine the results demonstrated in this paper through reproduction of their experiments.

# 2 Scope of reproducibility

This paper describes our efforts to reproduce the work from the paper Strategic Classification Made Practical[3], which addresses the problem of strategic classification in a manner that is more practical than previous approaches, more flexible than previous approaches and takes social good into account.

The original paper describes strategic classification as a classification task on a set of points $x \in \mathbb{R}^D$. A classifier $h(x)$ is tasked to classify $x$ to into classes $y = \{-1, 1\}$, which is determined by a score function $f$ via the decision rule $h(x) = \text{sign}(f(x))$. In strategic classification, the assumption is made that points can move according to a cost function $c(x, x')$. Therefore, users can modify their original features $x$ to their repsonse $x'$ to improve their outcome using the best move for $x$; $\Delta_h(x)$. Where this best move $\Delta_h(x)$ can be described as follows:

$$\Delta_h(x) = \underset{x' \in \chi}{\arg \max} \, h(x') - c(x, x') \tag{1}$$

To accurately classify points that are able to to modify their features, the error function has to take this into account. This results in an empirical loss function of:

$$\min_{f \in F} \sum_{i=1}^{m} \mathbb{I}\{y \neq h(\Delta_f(x))\} + \lambda R(f) \tag{2}$$

Which translates to choosing a score function $f$ such that misclassification of manipulated datapoints is minimized according to regularization method $R$ and $\lambda$ that determines the regularization strength. Optimizing equation 2 is referred to as strategic empirical risk minimization (SERM) in the original paper, which functions as the main loss function of the framework.

The claims that the paper made are as follows:

1. Flexible, practical and effective modeling: By using SERM, the claim is made that the framework can extend beyond the original formulation of strategic classification, i.e. outperform the original paper by Hardt et al.[2] and demonstrate good performance in new and realistic environments.

2. Socially-aware learning: By regularizing based on social objectives such as expected utility, social burden and recourse, the claim is made that the model can promote socially favourable outcomes, i.e. increase positive user outcomes as regularization increases.

In addition to reproducing the results presented in the paper, we perform novel experiments that test the claimed practicality, effectiveness and flexibility of the approach. In the experiments, we lift the assumption that all the users of the system are modifying their features to game the classifier. We make the case that this assumption cannot be applied to many real life settings. The experimental results show that in some settings the proposed method leads to decreased performance.

# 3 Methodology

## 3.1 Model descriptions

The focus of the original paper is on proposing a new framework, rather than a new model. Therefore, the models that were used in the original paper were relatively simple, consisting of linear classifiers and a basic RNN. All models were optimized using Adam.

The claims made in the paper were proven independently of each other in different experiments using different datasets, *spam, credit, fraud* and *financial distress* (datasets will be elaborated upon in section 3.2). This was done to demonstrate the flexibility of the framework as well as to demonstrate the generalizability of the framework on different datasets.

To verify said generalizability and flexibility, we repeated these experiments using the code provided by the authors. Similar to the original paper, results were compared to two "simpler", non-strategic models to verify performance. By non-strategic, we mean a standard classifier which assumes that the datapoints cannot move in any way. These models are referred to as **benchmark**, which is a non-strategic classifier trained and evaluated on non-strategic data, and **blind**, which is a non-strategic classifier evaluated on strategic data. To clarify: performing strategic classification is the act of training a model on a dataset and consequently performing a classification task where strategic movement of points according to a cost function is taken into account.

In the original paper, the model calculates the cost depending on the scale. Therefore, we will also be using scale to calculate the cost. Similar to the original paper, the influence of scale on model performance will be examined, where scale will take on the following values: $[0.5, 1, 2]$. These values were chosen such that the results can be compared to the original paper. This means that for each run of the model the cost is increased or decreased depending on the scale. The cost function used is:

$$Cost = scale * squared\_error \tag{3}$$

### 3.1.1 Experimental setup claim 1

The claim of model flexibility is tested by performing strategic classification on the *spam* dataset, which is the same dataset used by Hardt et al[2] in their original definition of strategic classification. The model used for this experiment is a linear classifier with a SERM loss function. Average accuracy was monitored and compared to a blind model and a benchmark model. To further validate the performance of the framework, strategic classification was also performed on the remaining datasets.

### 3.1.2 Experimental setup claim 2

The claim of socially-aware learning was verified by performing strategic classification on the *credit* dataset. To account for social good, several regularization techniques are used. Specifically expected utility, social burden and recourse are considered in the loss function:

Expected utility: summed utility that the users gain from classification results, minus the total cost of gaming

$$R_{utility} = -\sum_{i=1}^{m} h(\Delta(x)) - c(x, \Delta(x)) \tag{4}$$

Social burden: minimal cost value from among users classified as positive.

$$R_{burden} = \sum \min c(x, \Delta(x)) \tag{5}$$

Recourse: the capacity of a user who classified negative to restore approval through reasonable action

$$R_{recourse} = \sum_{i=1}^{m} \sigma(-f(x)) * \sigma(-f(\Delta_f(x))) \tag{6}$$

Where $\Delta(x)$ is the best response for $x$, $\Delta_f(x)$ is the best response for $x$ with regard to $f$, $x'$ is the strategically modified datapoint (as described in section 2), and $\sigma(x)$ is the sigmoid function of $x$. Similar to the first experiment, this

experiment uses a linear classifier with a SERM loss function. For every regularisation method, differing ranges of regularisation were analysed and compared to a benchmark model. The experiment on utility used a log range for $\lambda \in [-0.4, -0.2]$ of 10 steps, the experiment on social burden used a log range for $\lambda \in [-2, 1]$ of 30 steps, and the experiment on recourse used a log range for $\lambda \in [0, 0.3]$ of 15 steps.

### 3.1.3 Additional experiments

The authors make an assumption that all the users of the system will game according to their cost function. However, in many real life situations this assumption may not hold. For instance, in an email spam classification setting, people who write regular non-spam emails will most likely not think about gaming the spam classifier system. Assuming that every user is gaming might lead to a situation where a non-gaming email is falsely classified as spam due to the classifier being too strict. The case in which the strategic model is evaluated on a dataset consisting of non-gaming users is not taken into account in the paper.

We performed an additional experiment to investigate that case, with the hypothesis that the strategic model would perform worse on non-strategic data and would result in more false negative errors. In addition, we investigated a mixed data situation, in which gaming users make up some proportion of the data, defined by an additional parameter. After being trained on the original dataset, the classifiers were evaluated on $n$ datapoints, out of which $\beta * n$ points chosen at random were strategically modified. $\beta$ is the share of gaming users in the data, ranging from $[0, 1]$ in steps of 0.1. In this experiment, we tracked the classification accuracy and error types, false negatives and false positives, of the strategic and non-strategic model. The accuracy benchmark used with this model is the performance of the non-strategic model on non-strategic data ($\beta = 0$), which is the same as in the original paper. The experiment was performed on the four datasets from the paper in a new Jupyter notebook. We used the cost scale of 0.5 in the experiments.

### 3.2 Datasets

The datasets that were used in the original paper were also used in this work. All datasets and their details are shown in figure 1. *Spam*, as used by Hardt et al[2] contains features of users and spammers from a Brazilian social network. The features consists of numerical values about the user and their activity, such as amount of followers or number of words in a post. The dataset can be obtained from Costa et al.[1]. *Financial Distress*, created for a Kaggle challenge [5] contains time-series data describing the measure of financial distress for 422 companies. Each company has a maxium of 14 time steps after which the company has or has not gone bankrupt. *Fraud*, which was also created for a Kaggle challenge[4], contains 284000 credit card transactions that are either real or fraudulent. Features include numerical features related to the transaction, such as time of transaction and amount of money in the transaction. *Credit*, created by Ustun et al[6], contains credit card spending patterns as well as labels that define if the pattern is regular or not. There are 30000 data points, each with 11 features that include features such as age, payment history and education level.

|  | Dataset size | Features | Format | Description |
|---|---|---|---|---|
| Spam | 7,076 | 15 | .csv | Collection of social network users and posts |
| Credit | 30,000 | 11 | .csv | Collection of credit card spending patterns |
| Fraud | 284k | 29 | .csv | Collection of credit card transaction |
| Financial distress | 422 | 83 | time series | Collection of financial situations of companies |

Figure 1: Description of the datasets that were used in this project. All of these datasets have previously been used in the context of strategic classification, which makes them suitable to use when comparing models. Each dataset was gathered from the respective paper that they were first mentioned in.

### 3.3 Hyperparameters

As described in section 3.1, a seperate model was trained for every experiment. In turn, every model had seperate hyperparameters as well. As such, hyperparameters were chosen based on their performance in the original paper. Figure 2 and figure 8 (see Appendix B) show the hyperparameters that were used per experiment.

|  | Vanilla | | | |
|---|---|---|---|---|
|  | Spam | Credit | Fraud | Distress |
| Epochs | 16 | 16 | 16 | 16 |
| Learning rate | 0.5 | 0.5 | 0.5 | 0.5 |
| Batch size | 128 | 64 | 24 | 24 |
| Train slope | 1 | 1 | 1 | 1 |
| Eval slope | 5 | 5 | 3 | 5 |

Figure 2: Hyperparameter details for strategic classification without extensions (referred to as vanilla) on all datasets. Hyperparameters were chosen based on their performance in the original paper.

## 3.4 Experimental setup and code

The reproduction of the results was done with jupyter notebooks, this is the same as the original authors. Each notebook specialises in one part of the methods used in the original paper. The results of these different notebooks are then summarised in a plotting notebook called "reproduction plots.ipynb". Each notebook trains the respective model and saves the results to a .csv file. The measure used to evaluate the experiments is the accuracy of the model with the different datasets. The code can be found at this **github**.

## 3.5 Computational requirements

Experiments were able to be reproduced on laptops without a dedicated graphics card. Times per experiment differed, but individual experiments generally took 60 minutes. The exception to this rule was calculating the performance of the model for social burden, which took around 9 hours to run on a laptop with a GTX 1050.

# 4 Results

The reproduction study reveals minimal differences between the reported and reproduced accuracies. The first claim for a novel learning framework for strategic classification is supported by our reproduced results. The framework performs better than the Hardt et al.[2] strategic classification baseline. Thus, it also supports the second claim of the original paper.

## 4.1 Results reproducing original paper

| Dataset | Accuracy | | | | | | |
|---|---|---|---|---|---|---|---|
| Cost scale | | 0.5 | | 1 | | 2 | |
|  | Method | Original | Rerun | Original | Rerun | Original | Rerun |
| | Benchmark | 0.738 | 0.738 | 0.736 | 0.738 | 0.736 | 0.738 |
| Credit | SERM | 0.72 | 0.738 | 0.736 | 0.732 | 0.735 | 0.733 |
| | Blind | 0.57 | 0.55 | 0.625 | 0.587 | 0.665 | 0.647 |
| | Benchmark | 0.928 | 0.917 | 0.928 | 0.917 | 0.94 | 0.917 |
| Distress | SERM | 0.916 | 0.917 | 0.916 | 0.917 | 0.928 | 0.905 |
| | Blind | 0.63 | 0.631 | 0.642 | 0.631 | 0.666 | 0.643 |
| | Benchmark | 0.949 | 0.959 | 0.954 | 0.959 | 0.964 | **0.959** |
| Fraud | SERM | **0.76** | 0.908 | 0.954 | 0.908 | 0.939 | **0.719** |
| | Blind | 0.653 | 0.668 | 0.668 | 0.724 | 0.760 | 0.765 |
| | Benchmark | 0.814 | 0.813 | 0.815 | 0.813 | 0.819 | 0.813 |
| Spam | SERM | 0.719 | 0.797 | 0.787 | 0.804 | 0.779 | 0.794 |
| | Blind | 0.653 | 0.668 | 0.588 | 0.581 | 0.617 | 0.601 |

Figure 3: Accuracy table for the basic framework experiments that were reproduced, for all original datasets. Various scales (amount of gaming per user) were examined and results are generally very similar.

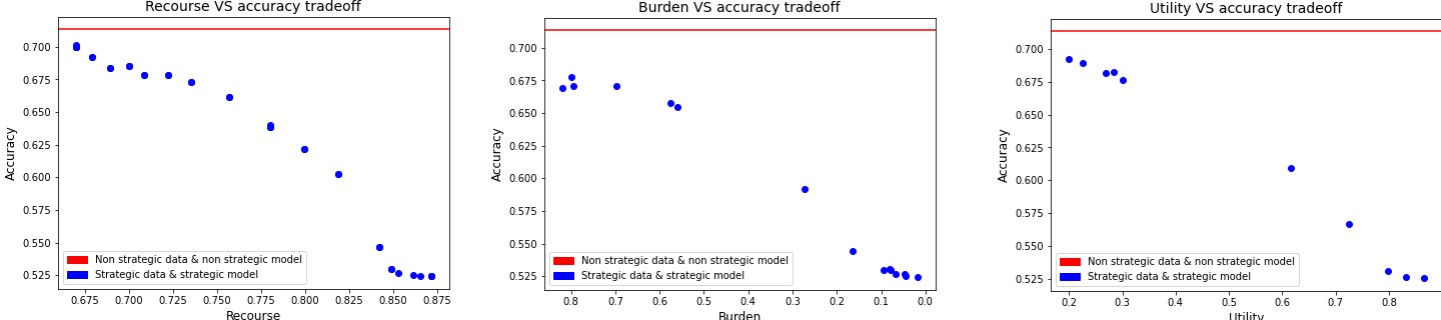

Figure 4: Graphs showing the relationship between the social welfare metrics and model accuracy (left: recourse, center: social burden, right: utility). Points correspond to different degrees of regularization $\lambda$.

### 4.1.1 Result 1

As mentioned in section 3.1, the claim of flexible modeling was reproduced and evaluated by running the code that was provided by the original authors and comparing the outcome to the results produced by Hardt et al.[2], as visible in figure 3 and appendix A. Similar to the results proposed in the original paper, our reproduced results outperform the algorithm proposed by Hardt et al. [2].

To further evaluate the claim of flexibility, the original paper examines performance in environments other than the one proposed in Hardt et al.[2]. The new environments are the datasets credit, financial distress and spam. These datasets give an accuracy of more than 0.7 for all except for the blind testing. This is true in both the original results and our reproduction. The lower accuracy in the blind testing can be explained by making the assumption that nobody is gaming when everyone is gaming in the dataset. This causes the agents who are gaming to cross the decision boundary by gaming the system. However, as visible in figure 3, the results of our reproduction of strategic classification on the *fraud* dataset using the SERM method has a significantly higher accuracy than the original paper. The other noticeable result in the accuracy is again in the fraud dataset. However, it is between the benchmark and the SERM method which in all other cases had a difference of less than 0.1.

### 4.1.2 Result 2

To prove the second claim about social impact, the authors made plots similar to figure 4. In figure 4 we see that there is an initial range where the accuracy doesn't drop significantly with increase in regularisation, similarly to the plots of the original authors. This means that the model can be fitted to accomplish lower social burden without having a detrimental effect to the accuracy. To test the true social impact it would be good to also test these regularized models on mixed data, since we saw a large amount of false negatives in the previous section. However we did not have time for that in this paper and these results show that the model can be regularized succesfully which was the main focus of the claim made by the original authors.

## 4.2 Results beyond original paper

The results in terms of accuracy, presented in Figure 5, show that for 3 out of 4 datasets, there is a clear linear relationship between the amount of gaming users and both models' performance, with roughly the same slope and opposite direction. For fraud and spam data, the point of equal performance is around 0.6, which suggests that the non-strategic model is better, assuming uniform probability distribution of $\beta$. The plots of false positive and negative errors (Fig 6) show, in line with our initial hypothesis, that the drop in performance of the strategic model is caused by more false negative errors, and by false positive in case of the non-strategic model.

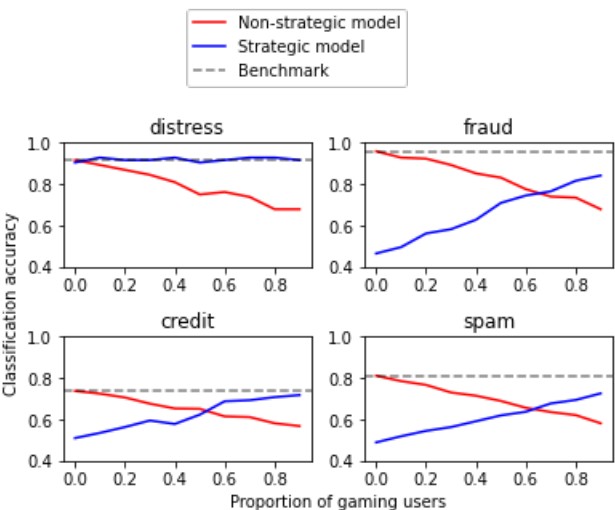

Figure 5: A comparison of non-strategic and strategic model accuracy as the proportion of gaming users changes, evaluated on four datasets used in the original paper.

The distress data is an outlier, in which the strategic model is at the benchmark level for all levels of $\beta$ and the non-strategic model's accuracy significantly decreases with $\beta$. This might be due to the fact that the distress classifier is a RNN network, while the other ones are linear models. The RNN can learn a more complex representation, which enables the strategic model to adjust in a more refined way, not just by moving the linear boundary.

The experimental results show a practical limitation in the presented method, which comes from the assumption that all users game in the same way and according to the same cost function. However, the distress model shows that decreased performance for non-strategic users is not always the case. Looking into this relationship is a potential direction of new research, which might result in improvements to the proposed framework.

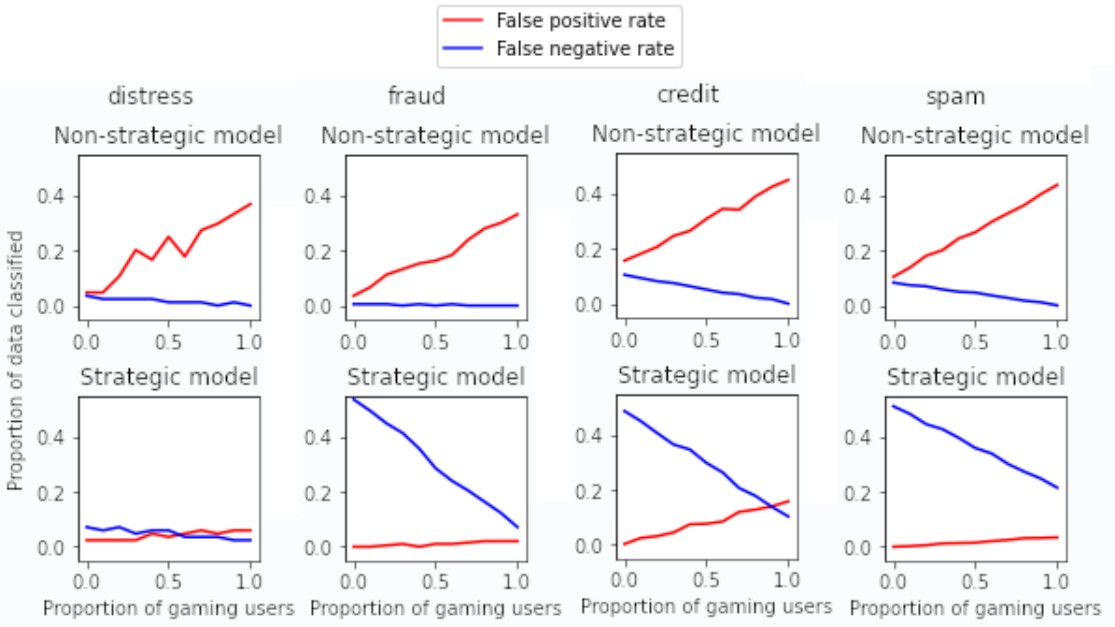

Figure 6: Fractions of different datasets classified as false negative and false positive by the non-strategic (top row) and strategic (bottom row) models. Evaluated on four datasets used in the original paper.

# 5   Discussion

After comparing the results, we conclude that the experimental results support the first claim to a large extent. The model is flexible since it performs well on multiple problems and it is effective since it performs significantly better on strategic data than the non-strategic model. On the practical part there are still some unanswered questions. As mentioned we thought that the original test environment with the assumption that all users are gaming might not be very representative for certain problems, like the spam dataset. We think most people don't adjust their emails to account for spam filters. We see this in the amount of false negatives from the strategic model on mixed data. For other problems like the distress dataset the strategic model performs really well on our mixed data, showing great potential for real-life problems. Follow-up research might involve investigating how different types of non-linear classifiers (e.g. decision trees, different neural network architectures) deal with the problem of mixed data.

The second claim is also supported by the results, they show that the model can be regularized without completely compromising the accuracy. This is an interesting result and it shows that their technique does work in practice. However since the strategic models show more false negatives for a mixed data set than non-strategic ones, it might have a more severe social burden than initially thought. An interesting followup experiment could be to check a regularized model on mixed data to see what the impact actually is.

## 5.1   What was easy

Since we had not worked on the problem of strategic classification before, it would have been very difficult to implement it in the limited timeframe that was available. Luckily, the authors of the paper had created a codebase containing all experiments and results. Along with the code, the dataset sources were also mentioned in the original paper. Therefore, we were able to verify that the results from the original paper were valid as soon as we got the code working.

## 5.2   What was difficult

Although the code was available, documentation of the code was quite sparse and unclear, and getting every part of the code to run took some trial and error due to the lack of comments. There were some short comments in the vanilla notebook, which we could consequently use to try to understand what was happening in the code. As well as a lack of documentation in the code, the counterpart of equations 1 and 2 written in the original paper lacked sufficient explanation about what the variables/functions meant and what their purpose was in the equation. When taking all these factors into account, understanding the code was quite a challenge.

Another problem was the fact that the results from the original paper were not summarised in a table, which meant we had to manually keep track of the accuracies for every experiment. Another problem was the fact that hyperparameter selection also was not elaborated upon. Although these are minor issues, they still took time to look into and were part of what was difficult about this project.

## 5.3   Communication with original authors

The communication with the authors was minimal. We asked about a clarification point about their method that had to do with them not using a combined dataset where users where partially gaming the system.

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

## A   Reproduced plots

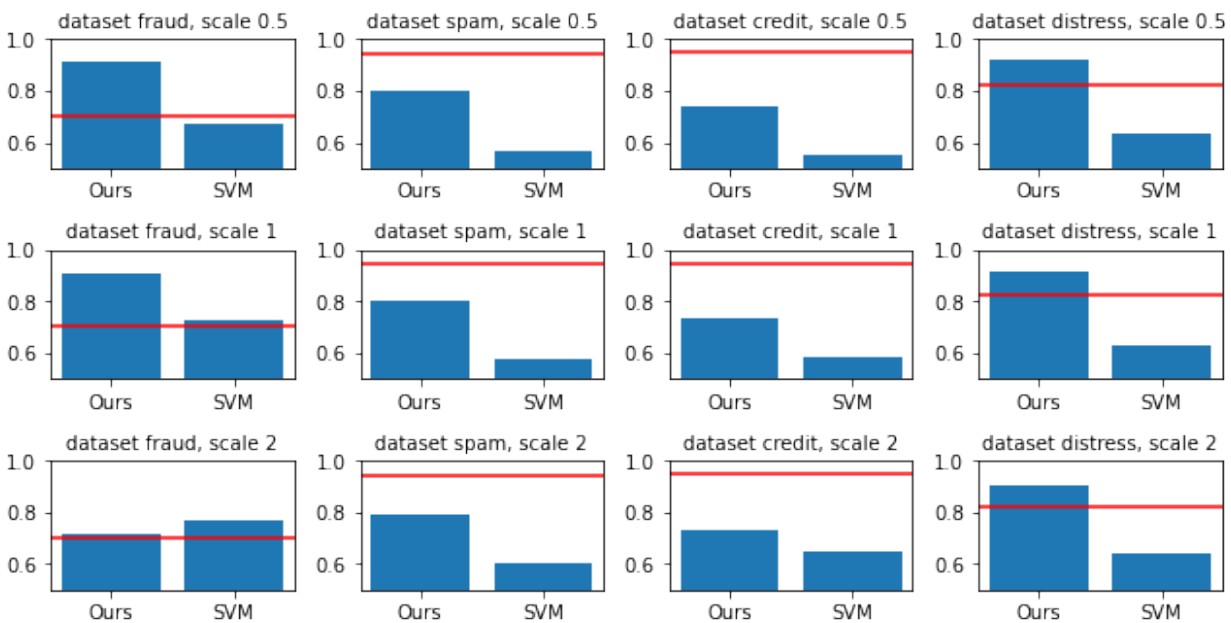

Figure 7: Graph detailing performance of the proposed method (SERM) on the different datasets for different amount of gaming compared to an SVM classifier.

## B   Additional information

|  | Social Regularization on *credit* | | |
|---|---|---|---|
|  | Utility | Recourse | Social burden |
| Epochs | 10 | 10 | 10 |
| Learning rate | 0.05 | 0.05 | 0.05 |
| Batch size | 64 | 64 | 64 |
| Cost scale | 1/x_dim | 1/x_dim | 1/x_dim |

Figure 8: Hyperparameter details for strategic classification with social regularization on the *spam* dataset.

