# OpenReview forum: "Strategic classification made practical: reproduction"
_ML_Reproducibility_Challenge/2021/Fall — RC2021 OutstandingPaper_

### Official Review · Reviewer_FE1r · 2022-03-01

**Rating:** 8
**Confidence:** 4

**Review:**

The authors provided a clear and concise reproducibility summary that illustrate some of the interesting results that follow. The paper is generally well structured and well written, with exception of the titles of some sections that are too generic and uninformative, such as "4.1.1 Result 1".

The authors also performed a hyperparameter search with the code that was already available. It would have been interesting to have more information on the "results 2" section. The section that stands out is 4.2 which shows the trade-off between a strategic and non-strategic model for different distributions of the dataset. It would be interesting to investigate this further, perhaps by letting the model estimate through the data which kind of dataset is it being faced with, and according choose whether to use a strategic or non-strategic model.


Minor:

-all the tables are treated as figures, which should not be the case.

-Equation 3 could be more clear, in part by not using text

---

### Official Review · Reviewer_Vfmh · 2022-03-01

**Rating:** 7
**Confidence:** 3

**Review:**

Overall, I think this is a fine reproduction study. Extending the originally proposed ideas to a new dataset is a good way to test the reproducibility and generalizability of the ideas. I also appreciate some of the care that went into the code notebooks created by the authors in their added notebook (i.e. "Reproduction plots.ipynb"). The added context for each cell adds a lot of readability to the code, and is a really useful artifact for others trying to reproduce this study. There are a few minor suggestions (see below) but I think the report should be accepted.

Suggestions:
- Did the authors respond to the questions you asked? It is unclear from your report.
- There are a few places where the formatting/writing needs to be addressed:
   - Section 2 first paragraph. The formatting is a bit off. Also the first sentence is a run-on and quite hard to parse at first read.
   - Section 4.1.1: Again the first paragraph has some odd formatting going on.
  I would also just go over the document a few times for editing.

---

### Official Review · Reviewer_PqFm · 2022-03-07
**Limited replication, but interesting extension**

**Rating:** 6
**Confidence:** 4

**Review:**

This paper reproduces “Strategic Classification Made Practical” by Levanon and Rosenfeld (2021; arXiv 2103.01826v2). The original paper proposes a framework for modeling “Strategic Classification”, a setting where motivated agents can selectively modify their submissions to a classifier so as to improve their odds of obtaining a desired outcome. For example, a business might tweak its application for a loan so that it matches the bank’s decision rule more precisely.
A major contribution of the original paper is the development of a conceptual framework, but it also includes some test-cases.

In the first part of this replication study, the authors reproduce those results. The authors used the original code (Section 4.1.1 “by running the code that was provided by the original authors”) on the same data set with the same hyper parameters. As a result, one would expect close agreement between the two studies. This was generally the case, but there were two large differences in accuracy (Figure 3: Fraud Dataset; SERM):
	* Cost = 0.5: Original value 0.76; Re-run=0.91
	* Cost = 2.0: Original value = 0.939; Return =0.719

I was disappointed that these were not discussed or investigated in the text, as it seems to undercut the claims that there were minimal differences between the reported and reproduced accuracies. The second part of the replication (on “social burden”) seems okay.

In the second section, the authors extend the model to consider cases where some, but not all, of the submissions attempt to strategically alter their model. This is an interesting and novel contribution, and seems like it would improve the real-world applicability of this framework. I would almost like to see this developed further as an independent paper, especially since the other replication parts seem somewhat underwhelming (no hyper parameter sweep, no investigation of the discrepancies), etc.

---

### Meta-Review · Area_Chair_zo9x · 2022-04-09

**Recommendation:** Accept (Outstanding Paper)
**Confidence:** 4

**Metareview:**

Reviewers praised the quality of the reproduction, especially the extensions to the original work.

---

### Decision · Program_Chairs · 2022-04-09

**Decision:**

Accept (Outstanding Paper)

**Comment:**

Following the recommendation of reviewers and meta-reviewer, the paper is accepted for ML Reproducibility Challenge 2021, and will be published in the upcoming special edition of ReScience Journal.

Additionally, after several rounds of discussion and incorporating recommendations from the Area Chairs and Program Chairs, the report has been granted an **Outstanding Paper Award** due to its exceptional quality of all-round reproducibility effort. Congratulations!